# Research on Precipitation Forecast Based on LSTM–CP Combined Model

**Yan Guo** [1,2], **Wei Tang** [1,2], **Guanghua Hou** [1], **Fei Pan** [1], **Yubo Wang** [1] **and Wei Wang** [3,*]

1 College of Information Engineering, Sichuan Agricultural University, Ya'an 625000, China; 14403@sicau.edu.cn (Y.G.); sau_tangwei@126.com (W.T.); 201803671@stu.sicau.edu.cn (G.H.); fei.pan@sicau.edu.cn (F.P.); 201902255@stu.sicau.edu.cn (Y.W.)
2 Key Laboratory of Agricultural Information Engineering of Sichuan Province, Sichuan Agricultural University, Ya'an 625000, China
3 College of Management, Sichuan Agricultural University, Ya'an 625000, China
* Correspondence: wangwei@sicau.edu.cn

**Abstract:** The tremendous progress made in the field of deep learning allows us to accurately predict precipitation and avoid major and long-term disruptions to the entire socio-economic system caused by floods. This paper presents an LSTM–CP combined model formed by the Long Short-Term Memory (LSTM) network and Chebyshev polynomial (CP) as applied to the precipitation forecast of Yibin City. Firstly, the data are fed into the LSTM network to extract the time-series features. Then, the sequence features obtained are input into the BP (Back Propagation) neural network with CP as the excitation function. Finally, the prediction results are obtained. By theoretical analysis and experimental comparison, the LSTM–CP combined model proposed in this paper has fewer parameters, shorter running time, and relatively smaller prediction error than the LSTM network. Meanwhile, compared with the SVR model, ARIMA model, and MLP model, the prediction accuracy of the LSTM–CP combination model is significantly improved, which can aid relevant departments in making disaster response measures in advance to reduce disaster losses and promote sustainable development by providing them data support.

**Keywords:** precipitation forecast; long short-term memory network; Chebyshev polynomial; BP neural network



## 1. Introduction

Disasters caused by natural hazards can often lead to significant and long-lasting disruptions of the whole socioeconomic system. One catastrophic event, such as a flood, can destroy multi-infrastructure systems, lead to cascading failures and substantial socioeconomic damages, and hinder development. A large amount of precipitation will directly lead to floods and waterlogging disasters and make crops impossible to harvest, as well as easily cause secondary disasters [1], such as collapses, landslides, mudslides, and waterlogging. The causes of precipitation are highly complex [2–4] due to the comprehensive influence of monsoons, topography, urban distribution, temperature, and evaporation, leading to more difficulties in predicting precipitation. In addition, rainfall also has some fixed characteristics, and its influencing factors, such as terrain, urban distribution, and temperature, will not change greatly in a short time. Precipitation also shows a high degree of regularity.

With the continuous progress of technology, artificial intelligence (AI) has become an important driving force in various fields, including sustainable development. Deep learning can improve the ability to deal with complex problems and help us increase our understanding of variables and sources that affect rainfall. At present, there is a myriad of existing studies on precipitation prediction, among which forecasts based on regression analysis and forecasts based on time series are two classic forecasting approaches.

Forecasts based on regression analysis mainly include autoregressive models, moving average models, autoregressive moving average models, and differential autoregressive moving average models [5]. Prediction methods based on time series can be mainly divided into grey systems [6], Markov models [7], and set pair analysis [8]. These methods are simple and widely used, but the accuracy of precipitation prediction is low, which cannot accurately describe the trend of precipitation development and change. With the rapid improvement of computer computing power and the development of big data [9], deep learning technology has become more and more widely used in recent years [10]. For one thing, deep learning is highly suitable for processing multi-dimensional and complex data, with no requirement for the physical modeling [11] of data; for another, deep learning has multiple levels, where low-level features are combined to form more abstract high-level features, and nonlinear network structure can achieve complex function approximation, showing powerful dataset representation capabilities. Therefore, using deep learning technology to predict precipitation has become a very practical value and challenging problem.

Among numerous deep learning technologies, BP and LSTM are two widely used deep learning neural networks [12,13]. The neural network has been put to work in many ways, including fitting, classification, and pattern recognition, since the BP algorithm was proposed [14]. For example, Ferreira et al. [15] evaluated the potential of deep learning and traditional machine learning models to predict daily reference evapotranspiration (seven days). The results show that the performance of the deep learning model is slightly better than that of the machine learning model. Granata et al. [16] established three models based on a recurrent neural network to predict short-term future actual evapotranspiration. The results show that the model based on deep learning can predict the actual evapotranspiration very accurately, but the performance of the model will be significantly affected by the local climate conditions. There is a myriad of improvements in BP neural networks made by researchers, one of which is to change the excitation function of the BP neural network. For example, Zhang et al. [17] took the sine function as the excitation function of the BP neural network. CP is a set of orthogonal polynomials that is often used for function approximation. Previous studies have shown that orthogonal polynomials perform better in fitting functions, and in comparison to ordinary polynomials [18–20], orthogonal polynomials have better fitting stability and fitting ability. CP already has a wide range of applications in neural networks. Zhang et al. [21,22] proposed a variety of neural network structures for classification, achieved by applying CP in a feedforward neural network and combing with the direct weight determination method, as well as the cross-validation method. Based on Zhang's research, Jin et al. [23,24] further improved the research as applied to wine region classification and breast cancer classification, respectively, and achieved good classification results. Unlike the BP neural network, the recurrent neural network (RNN) is a network dedicated to processing sequence data. The original RNN has poor processing capacity for sequence data due to its limited memory capacity, such that many improvements have been made on RNN by researchers. LSTM [25] is the most widely used network among many variants of RNN, with its ability to effectively alleviate the disadvantages of RNN, such as gradient disappearance and weak memory ability, making RNN widely applied in various fields. For example, Kratzert et al. [26] explored the potential of using a long-term and short-term memory network (LSTM) to simulate meteorological observation runoff, and verified by practice that its prediction accuracy is comparable to that of the perfect baseline hydrological model. Xiang et al. [27] used the prediction model based on LSTM and seq2seq structure to predict hourly rainfall runoff. The results show that the prediction accuracy of the LSTM-seq2seq model is higher than that of other models such as ordinary LSTM. This method is used to improve the accuracy of short-term flood prediction.

At present, researchers have applied the above two kinds of neural networks to the prediction of precipitation. The prediction approach of the neural network can effectively extract the random characteristics of a nonlinear sequence, which achieves a high prediction precision and has good research and application value. For example, according to the

meteorological data of Jingdezhen from 2008 to 2018, J. Kang et al. used the long-term and short-term memory neural network (LSTM) model to predict precipitation. The experimental results show that the LSTM model can be well applied to precipitation prediction [28]. Y. Zhou [29] used an improved BP neural network model to predict typhoon precipitation and typhoon precipitation events. By analyzing the difference in candidate predictors between normal years and years with a large prediction error, this method proposed a new predictor for the BP model in each iteration, and the precipitation prediction accuracy was better than that of the original BP neural network. In addition to predicting precipitation through deep learning methods, precipitation can also be predicted through satellite cloud images and radar detection. For example, Zahraei et al. [30] introduced a pixel algorithm for short-term quantitative precipitation forecasting (SQPF) using radar rainfall data, and proposed a pixel-based nowcasting (PBN) algorithm, which uses a hierarchical grid tracking algorithm. The image captures the high-resolution advection of storms in space and time. The results show that the proposed algorithm can effectively track and predict severe storm events in the next few hours. Bowler et al. [31] proposed a new Gandalf system precipitation prediction scheme based on advection. The method does not need to divide the radar analysis into continuous rain areas (CRA) and uses smoothing constraints to diagnose the block advection velocity in rainfall analysis by using the idea of optical flow. This scheme is compared with the old Gandolf advection scheme based on CRA, and the new scheme performs better in cases related to severe floods and in a continuous validation period of 3 months. Pham et al. [32] compared several advanced artificial intelligence (AI) models for predicting daily precipitation, and the results showed that support vector machine is the best method for predicting precipitation, and it was also found to be the most robust and effective prediction model. Banadkooki et al. [33] applied the flow pattern optimization algorithm (FRA) to the optimization of the multilayer perceptron neural network (MLP) and support vector regression (SVR), and established the precipitation prediction model. The results show that the performance of the proposed MLP-FRA model is better than all other models and has a stronger rainfall prediction ability. Wang et al. [34] combined satellite and radar observation data, and through proper orthogonal decomposition and assimilation of the data, the effect of precipitation forecasting was improved.

There have been many studies on precipitation prediction from the perspective of relevant studies at home and abroad, and an excellent application of LSTM in the prediction of sequence data has been achieved. However, the LSTM network structure is more complicated, and the number of network unit parameters is relatively large. A slight increase in the network depth will lead to a rapid increase in the number of parameters. The huge amount of parameters increases the difficulty of calculation. For medium and large datasets, higher performance equipment is required to perform calculations [35]. In addition, although the LSTM network overcomes the problem of gradient disappearance to a certain extent, the memory function of the LSTM network still depends on the long sequence. When the sequence is too long, the problem of gradient disappearance may still occur, which greatly affects the performance of LSTM [36], and the gradient vanishing problem has not been completely solved. At the same time, the LSTM network training model is more complicated and the training time is longer [37].

Given the above situation, this paper proposes to combine the Long Short-Term Memory (LSTM) [38] network and the Chebyshev polynomial (CP) [39], aiming to form an LSTM–CP combined model for rainfall prediction. From a theoretical point of view, this model combines CP and LSTM networks for the first time. Firstly, the LSTM network is used to extract the time-series features in the original data. Then, the BP (Back Propagation) neural network [40] with CP as the activation function is used to process the time-series features. This approach can effectively reduce the number of parameters, with the premise of ensuring accuracy, and has stronger characterization capabilities for sequence data, which provides a new idea for researchers in the field of neural networks. In the prediction of rainfall using a machine learning algorithm, the ARIMA model has low accuracy in predicting non-stationary or fluctuating time series [41]. The number of parameters in

the SVR model is usually very large [42]. The MLP network needs a large number of patterns and iterations to realize effective learning so it needs more execution time [43]. Compared with these classical machine learning algorithms, the LSTM–CP combined model proposed in this paper has higher accuracy, fewer parameters, and faster operation speed in the prediction of precipitation. The prediction results of the model in monthly units are relatively accurate, basically reflecting the changing trend of precipitation. It is helpful to provide a data reference for areas prone to floods and drought disasters, as well as help relevant departments to prepare in advance, reducing local economic losses. The model is capable of shortening the running time more effectively when dealing with large and medium-sized datasets as it can effectively reduce the use of parameters, making the process of sequence data more efficient.

This article is structured as follows: Introduction, where the importance and necessity of accurate precipitation forecasts are addressed and the existing precipitation forecasting methods and the existing problems are listed. The method section gives a detailed introduction to the related models and theoretical methods used, and compares and analyzes the parameters of different models. In the experimental evaluation section, the prediction models of LSTM, LSTM–BP, and LSTM–CP are constructed, respectively, and the parameter setting process of the LSTM–CP combined model is elaborated. The experimental results show that, compared with the ordinary LSTM neural network model, the LSTM–CP combined model proposed in this paper has fewer parameters, shorter running time, and relatively smaller prediction error than the LSTM network. At the same time, this paper also compares the LSTM–CP combined model with the traditional rainfall prediction SVR model, ARIMA model, and MLP model, finding that the prediction accuracy of the LSTM–CP combined model is significantly improved. Finally, the discussion of results and conclusions is presented, showing the ability to predict precipitation through the LSTM–CP combination model.

## 2. Materials and Methods

### 2.1. LSTM–CP Combined Model Framework

The LSTM network has inherent advantages in processing sequential data on account of its powerful memory [44–46]. In this paper, the LSTM network is used as the basic network of sequence data prediction, with the BP neural network combined to use its excellent function fitting ability. We can obtain an LSTM–CP combination model by using CP to improve BP neural networks. CP and LSTM networks are combined for the first time in this model, where the LSTM network is first used to extract the time-series features in the original data. Then, the BP (Back Propagation) neural network of CP as the activation function is used to process the time-series features, with the specific process shown in Figure 1.

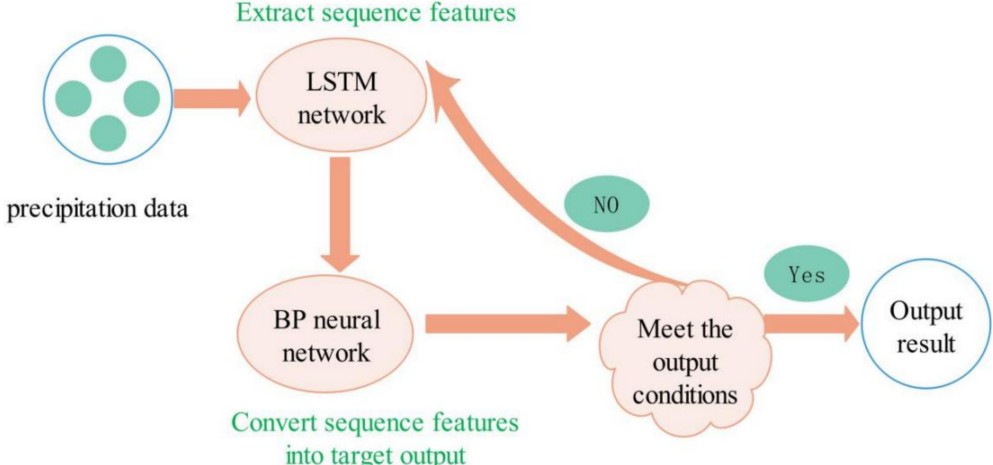

**Figure 1.** Combined model framework.

### 2.2. Feature Extraction Based on LSTM Network

Because the LSTM network has a strong memory capacity, it has a natural advantage in processing sequence data. This article employs the LSTM network as the basic network for sequence data prediction. In practical applications, RNN has been able to process some simple correlation information while its memory capacity is not strong. When the sequence is too long, error back propagation will cause larger gradient dispersion and gradient explosion problems, which can be effectively alleviated by introducing a "gate" mechanism [47,48] and memory unit [25,49] in the LSTM network.

#### 2.2.1. Basic Idea

Only two factors, the current round of input $x_t$ and the last round of output $h_{t-1}$, affect the traditional RNN network unit. Since there is only one tanh excitation unit in the network, the network output is:

$$h_t = \tanh(W_t[h_{t-1}, x_t] + b_t) \tag{1}$$

Therefore, RNN is sensitive to short-term input, making it difficult to solve the long sequence problem, as shown in Figure 2.

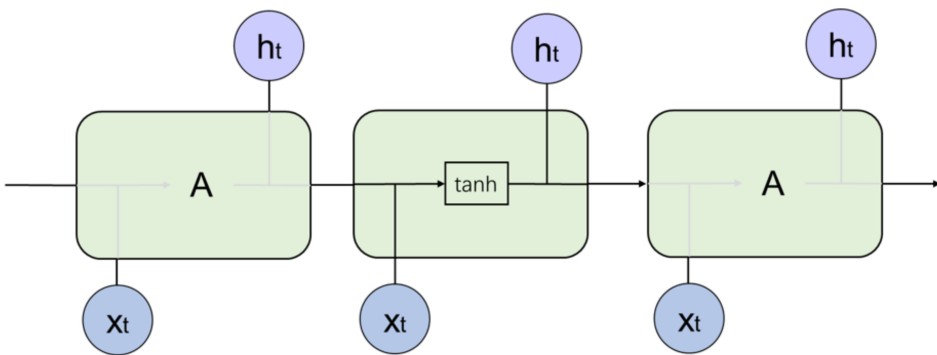

**Figure 2.** RNN network structure.

The LSTM network introduces a unit state and "gate" mechanism, which enhances the network's ability to remember long-term information, as shown in Figure 3. The current cell state $C_t$ consists of the previous cell state $C_{t-1}$, the previous cell output $h_{t-1}$, as well as the current input $x_t$. The forget gate and input gate process the output $h_{t-1}$ of the previous round and the input $x_t$ of the current unit, and then combine with the current unit state $C_t$ to form the output $h_t$ of the current round through the output gate.

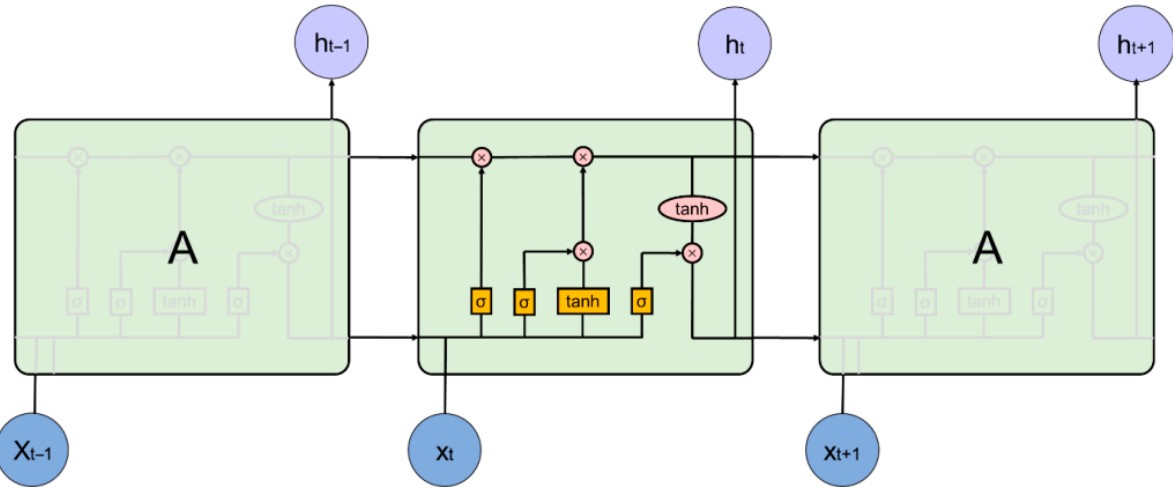

**Figure 3.** LSTM network structure.

It can be seen from Figure 3 that the unit state C, which runs through the whole LSTM network, constantly transfers information from the previous layer to the next layer, realizing the long-term memory retention function. In the LSTM network, there are three gate switches: input gate, forgetting gate, and output gate, through which the LSTM network can determine whether the current output depends on the early output, recent output, or current input.

### 2.2.2. Forgetting Gate

The first problem that the LSTM network solves is to determine the information that can pass through the current neuron, which is determined by the forgetting gate in LSTM. In the forgetting gate, the output $h_{t-1}$ at the previous moment is dot multiplied with the input $x_t$ at the current moment, and then the output $f_t$ at this moment inside the neuron is obtained through the Sigmoid function [50], which is:

$$f_t = \sigma\left(W_f[h_{t-1}, x_t] + b_f\right) \tag{2}$$

where $W_f$ represents the weight matrix and $b_f$ represents the bias term.

### 2.2.3. Input Gate and Unit Status

After confirming the reserved information, LSTM needs to determine how much of the current input needs to be stored in the cell state, with this function implemented by the input gate in LSTM. In the input gate, the current input $x_t$ together with the previous round of output $h_{t-1}$ are point multiplied and then passed through the function of Sigmoid, with the purpose to determine which inputs are updated; the current input $x_t$ and the previous round of output $h_{t-1}$ are subjected to a dot product operation and then passed through the tanh function, aiming to form alternative update information.

$$i_t = \sigma(W_i[h_{t-1}, x_t] + b_i). \tag{3}$$

$$C_t = \tanh(W_C[h_{t-1}, x_t] + b_C). \tag{4}$$

where $W_i$ and $W_C$ are the weight matrix, respectively, and $b_i$ and $b_C$ are the bias items, respectively. The current cell state $C_t$ is starting to be updated after obtaining the results of the forgetting gate and the input gate. The output $f_t$ of the current time in the neuron is point multiplied with the result $C_{t-1}$ of the previous round of the memory unit, while at the same time the two internal update information points $i_t$ and $\widetilde{C}_t$ perform the dot product operation, and finally the new unit state is obtained by adding them together.

$$C_t = f_t * C_{t-1} + i_t * C_t. \tag{5}$$

### 2.2.4. Output Gate

For LSTM, it is necessary to determine how to output the current information when the unit state is determined, and with this function determined by the output gate, the unit output is jointly determined by $x_t$, $h_{t-1}$, and $C_t$. $o_t$ is obtained through the function of the Sigmoid function after $x_t$ and $s_{t-1}$ are dot multiplied with $C_t$, which passes through the tanh function dot multiplied by $o_t$, and finally output $s_t$ is obtained:

$$o_t = \sigma(W_o[s_{t-1}, x_t] + b_o). \tag{6}$$

$$s_t = o_t * \tanh(C_t). \tag{7}$$

where $W_o$ is the weight matrix and $b_o$ is the offset term.

### 2.3. Convert Sequence Features into Target Output

2.3.1. CP Combined with BP Neural Network

The Chebyshev polynomial is an important special function named after the famous Russian mathematician Tschebyscheff. It originates from the cosine function of multiple angles and the expansion of the cosine function. It is divided into the first kind of Chebyshev polynomial and the second kind of Chebyshev polynomial. Chebyshev polynomials used in this paper belong to the first category. Chebyshev polynomials play a very important role in approximate calculation in mathematics, physics, and technical science, such as the injection continuous function approximation problem, impedance transformation problem, and so on. The roots of the first kind of Chebyshev polynomials (called Chebyshev nodes) can be used for polynomial interpolation. The corresponding interpolation polynomials can minimize the Runge phenomenon and provide the best uniform approximation of polynomials in continuous functions. In practical application, it is often necessary to solve a known complex function $f(x)$, and in order to simplify the calculation, it is usually necessary to find a function $Q_n(x)$ to minimize the error between the two in a certain metric sense. In the Chebyshev best uniform approximation theory, the function $Q_n(x)$ is a Chebyshev polynomial and it satisfies that the difference between and in an interval $[a, b]$ is the smallest of all polynomials $Q_n(x)$ and $f(x)$ in the interval, as shown in the following formula:

$$\max_{a \leq x \leq b} |Q_n(x) - f(x)| = \min \left| \max \left| Q(x) - f(x) \right| \right| \tag{8}$$

The function approximation theory of Chebyshev polynomials shows that such polynomials $Q_n(x)$ exist and are unique: let $D_x = \max\limits_{a \leq x \leq b} |Q_n(x) - f(x)|$, $D_x$ has at least $n + 2$ interleaving points $[x_1 \cdots x_{n+2}](a \leq x_1 < \cdots < x_{n+2} \leq b)$ on $[a, b]$, so that $D(x_i) = \pm D_n$, among them, $i \in [1, n + 2]$, $Q_n(x)$ is the best uniform approximation of $f(x)$.

Chebyshev polynomials are a series of orthogonal polynomials [51], which can approximate any continuous function. Neural networks based on CP have excellent capabilities in fitting as well as generalization. The Chebyshev polynomial is defined in a recursive manner, where CP can be expressed by the following recursive expression when the variable has a value range between $-1$ and $1$, for an $n$-th order CP:

$$T_0(x) = 1, \tag{9}$$

$$T_1(x) = x, \tag{10}$$

$$T_{n+1}(x) = 2xT_n(x) - T_{n-1}(x). \tag{11}$$

According to the theory of orthogonal polynomial approximation, a set of Chebyshev polynomials can approach any objective function, when the variables belong to $-1$ to $1$ and the number of polynomials $R$ is large enough. As follows:

$$f(x) \approx \sum_{r=0}^{R} w_r T_r(x). \tag{12}$$

where $R$ is the number of Chebyshev polynomials used to fit $f(x)$, $T_r(x)$ represents the $r$-th polynomial, and $w_r$ represents the weight of the $r$-th polynomial.

It can be seen from Equation (12) that the objective function $f(x)$ is obtained by the weighted sum of $R$ CPs. To express this more intuitively, this paper adopts the method of lexicographical sorting to express Chebyshev polynomials, and sorts them according to the order of each polynomial. Given two different basis functions $\varphi_q(x) = \mu_{i_1}(x_1) \ldots \mu_{i_N}(x_N)$ and $\varphi_{\hat{q}}(x) = \mu_{j_1}(x_1) \ldots \mu_{j_N}(x_N)$ in the condition of $q \neq \hat{q}$. Let $Q = [i_1, i_2, \ldots, i_N]$, $|Q| = [i_1 + i_2 + \ldots i_N]$, $\hat{Q} = [\hat{i}_1, \hat{i}_2, \ldots, \hat{i}_N]$, and $|\hat{Q}| = [\hat{i}_1 + \hat{i}_2 + \ldots \hat{i}_N]$, $q > \hat{q}$ is established when any of the following conditions are met:

Condition 1: $|Q| > |\hat{Q}|$;

Condition 2: $|Q| = |\hat{Q}|$, and the first non-0 element of $Q - \hat{Q} = \left[ i_1 - \hat{i}_1, i_2 - \hat{i}_2, \ldots i_N - \hat{i}_N \right]$ is positive.

The BP neural network usually consists of a myriad of layers, including one input layer, several hidden layers, and one output layer. It has already been proved that the BP neural network, with a single hidden layer, can approach any continuous function in the closed interval with arbitrary precision [52]. The BP neural network of a single hidden layer is combined with the LSTM network in this paper, and the topology diagram of the common single hidden layer BP neural network is shown in Figure 4.

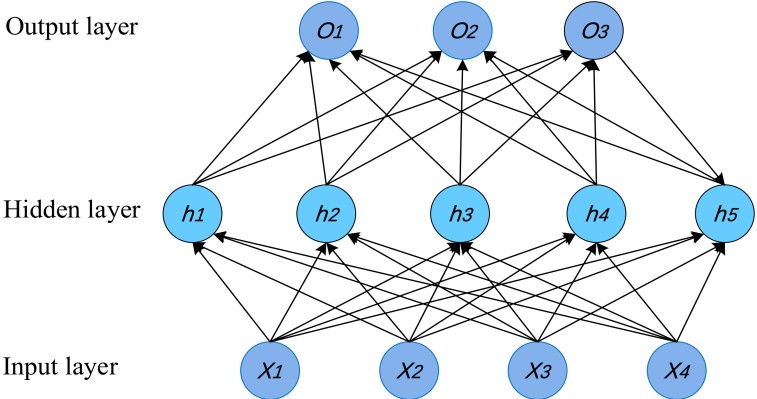

**Figure 4.** The topology of BP neural network.

In this network, where the input is $x_1 \ldots x_N$, the actual output is $o_1 \ldots o_k$ and the target output is $y_1 \ldots y_k$, each neuron in the input layer is fully connected with each neuron in the hidden layer, and each neuron in the hidden layer is fully connected with each neuron in the output layer. The weight between the $i$-th neuron in the input layer and the $j$-th neuron in the hidden layer is represented by $w_{ij}$, and the bias term is $a_j$. The weight between the $j$-th neuron in the hidden layer and the $k$-th neuron in the output layer is represented by $v_{jk}$, and the bias term is represented by $a_j$. The learning process of the BP neural network includes two steps, where the first step is the forward spread of information and the second step is the error back propagation. In the stage of the forward spread of information, information is transmitted forward, and the data are transferred from the input layer to the output layer through a weighted sum, with each neuron in the hidden layer and the output layer that can be, respectively, expressed as:

$$f(z_j) = f\left( \sum_{i=1}^{I} w_{ij} x_i - a_j \right), \tag{13}$$

$$o_k = f\left( \sum_{j=1}^{J} v_{jk} f(z_j) - \beta_k \right). \tag{14}$$

For the error back propagation stage, by computing the error and gradually correcting the weight and bias value through the gradient descent approach, the error and weight adjustment can be expressed as:

$$E = \frac{1}{2} \sum_{k=1}^{K} (o_k - y_k), \tag{15}$$

$$v_{jk} = v_{jk} - \eta \frac{\partial E}{\partial f(z_j)}, \tag{16}$$

$$v_{ij} = v_{ij} - \eta \frac{\partial E}{\partial f(z_j)} \times \frac{\partial f(z_j)}{\partial x_i}. \tag{17}$$

The "learning" process of the BP neural network is to gradually correct the weight and bias value according to the input data until the accuracy is satisfied or the maximum number of iterations is reached.

Hecht-Nielsen [52] has proved that a feedforward neural network with three layers can approximate any nonlinear continuous function in a closed interval with arbitrary precision. However, BP neural networks have some inherent shortcomings, such as slow convergence, ease of falling into a local minimum, and ease of falling into a saddle point, etc. The excitation function adopted by the traditional BP neural network is usually sigmoid, tanh, and ReLU, while this paper employs a set of linear independent orthogonal polynomials, which are Chebyshev polynomials instead of Sigmoid function, as the excitation function, as shown in Figure 5.

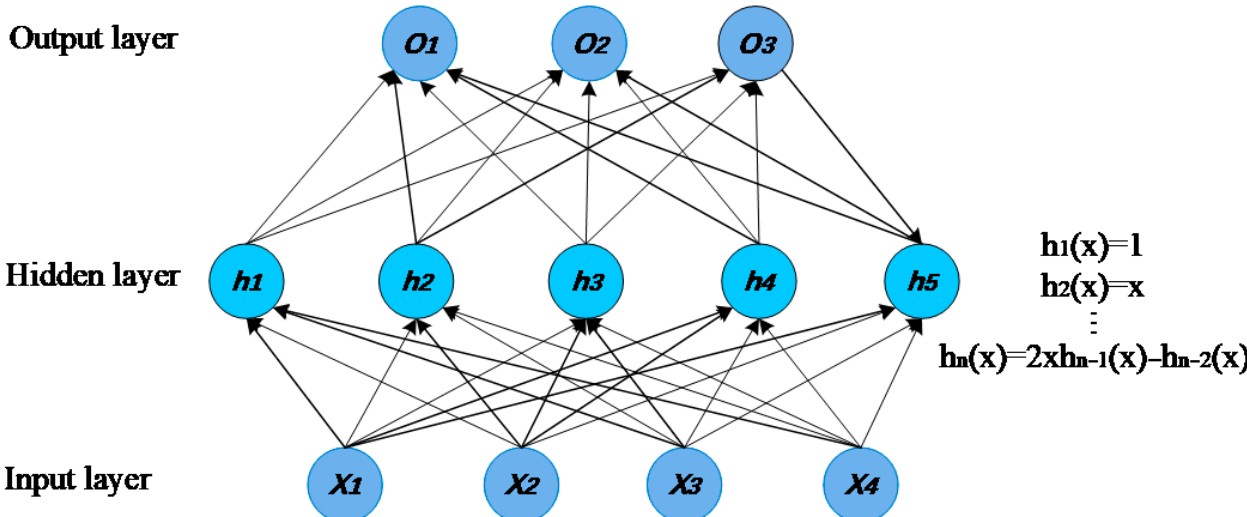

**Figure 5.** BP neural network based on CP.

A large amount of literature [17–19] has verified that using Chebyshev polynomials as the excitation function can effectively optimize the BP neural network. In the experiment of this paper, in contrast to networks and LSTM networks that use Sigmoid as the excitation function, the error of the network using CP as the excitation function declines faster and more steadily, and the prediction accuracy is also higher at the same time.

### 2.3.2. LSTM Combined with BP Neural Network

RNN is a typical feedback neural network whose network structure takes the time dimension into account, which can achieve excellent performance in processing data with timing laws. The structure of a single-layer RNN is shown in Figure 2, where each unit will receive the output of the previous unit and the input of this unit, and then the output can be given. Because the longer RNN is accompanied by the problems of gradient explosion and gradient disappearance, it has a limited memory capacity, which makes it unable to deal with long sequence data. In the actual operation of the LSTM network, the data need feeding into a linear layer to change the data dimension after passing through the LSTM network. This linear layer will transform the output of the LSTM network into a target output. Adopting the BP neural network to replace the linear layer of the LSTM network is considered in this paper, as shown in Figure 6.

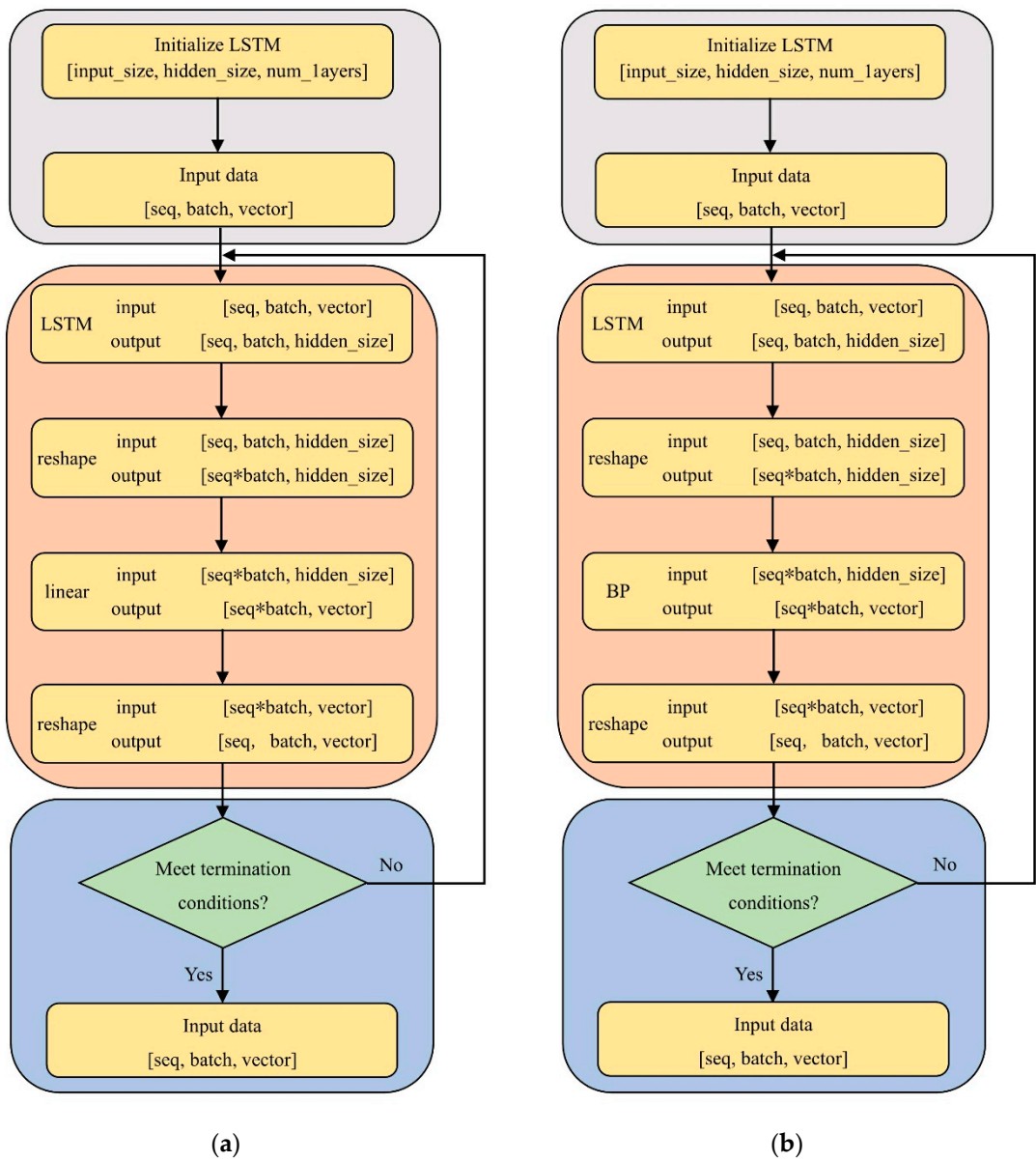

**Figure 6.** Network operation process. (**a**) LSTM operation process. (**b**) LSTM–BP operation process.

Firstly, the LSTM network is employed to process the original data, with the purpose to extract the time-series features of the data, and then the characteristic data are fed into the BP neural network with CP as the excitation function. In this way, the prediction ability of LSTM for sequence data and the function fitting ability of the BP neural network are used at the same time. This can effectively overcome the shortcomings of the BP neural network, such as slow convergence and ease of falling into a local minimum and local saddle point, through changing the excitation function of the BP neural network to CP.

*2.4. Parameter Analysis/Complexity Analysis*

The parameters of the LSTM–CP combined model consist of two parts: one is the parameters of the LSTM network and the other is the parameters of the BP neural network. Jin et al. [24] have proved that, for a fully connected feedforward neural network, the computational complexity of CP as an excitation function is lower than that of a Sigmoid. Therefore, this paper only focuses on the parameters of LSTM that combined with the BP neural network with CP as the excitation function.

The LSTM network has a total of three "gates" and a unit state, where each of the three gates generates some parameters. In contrast, the unit state does not generate any new parameters, and some parameters are also generated in the BP neural network. The number of parameters of different networks will be analyzed next according to the network operation flow chart in Figure 6.

For the forget gate, $h_{t-1}$ is the output at the previous moment and the length is $m$; $x_t$ is the input at the current moment and the length is $n$; $W_f$ represents the weight matrix and the matrix size is $[m+n, m]$; and $b_f$ represents the offset term and the length is $m$. For the input gate, $h_{t-1}$ is the output at the previous moment and the length is $m$; $x_t$ is the input at the current moment and the length is $n$; $W_i$ and $W_C$ are the weight matrices, whose matrix sizes are both $[m+n, m]$; and $b_i$ and $b_C$ are, respectively, offset terms and the length is $m$. For the output gate, $h_{t-1}$ is the output at the previous moment and the length is $m$; $x_t$ is the input at the current moment and the length is $n$; $W_o$ is the weight matrix and the matrix size is $[m+n, m]$; and $b_o$ is the bias term and the length is $m$. For the memory unit, it only performs a dot multiplication operation between the current output $f_t$ in the neuron and the last round of memory unit result $C_{t-1}$, and the two internal update information points $i_t$ and $\widetilde{C}_t$ perform the dot multiplication operation, with no new parameters generated. Therefore, the parameters of the forget gate, input gate, and output gate are, respectively:

$$s1 = (m+n) * m + m, \tag{18}$$

$$s2 = 2 * ((m+n) * m + m), \tag{19}$$

$$s3 = (m+n) * m + m. \tag{20}$$

When the number of LSTM network layers is Q, the total parameter quantity of the LSTM network is determined by the number of network layers, as well as the number of parameters of the three "gates", and the total parameter quantity of the LSTM network is:

$$s = Q_1 * 4 * ((m+n) * m + m). \tag{21}$$

In each round of parameter training, the parameters of the LSTM network and BP neural network will be updated at the same time, and the input of the BP neural network is determined by the LSTM network output. According to the above analysis, for the BP neural network, the input is $h_t$ and the length is $m$. Let the number of neurons in the BP network be $R$, the output be $h_t$, and the length be $m$. The LSTM–BP combined model in this paper adopts the BP neural network to replace the linear layer of the LSTM network, with CP as the excitation function of the BP neural network, and each neuron is fully connected to the output, as shown in Figure 5. Then, the number of parameters of the BP neural network is:

$$s4 = m * R + m * R = 2mR. \tag{22}$$

The number of parameters in the LSTM–CP combination model is:

$$S = s + s4 = Q_2 * 4 * ((m+n) * m + m) + 2mR. \tag{23}$$

In summary, the parameters of each network are listed in Table 1.

This paper studies the precipitation data of 784 months in Yibin City from 1951 to 2017. The precipitation data of 1971 are ignored due to the missing data from January to June in 1971. In each round, 90% of the data are selected for training, and then the length is $n = 703$. The experiment of Section 3.1 shows that in LSTM the length of $h_t$ is $m = 16$ and the number of LSTM network layers is $Q1 = 2$, and in LSTM–CP the number of LSTM network layers is $Q2 = 1$, the length of $h_t$ is $m = 32$, and the number of BP neural network neurons is $R = 6$. The various parts and overall parameters of the network used in this article are shown in Table 2.

**Table 1.** Network parameter quantity function.

| Structure | Parameter Quantity |
|-----------|--------------------|
| forgetting gate | $s1 = (m + n) * m + m$ |
| input gate | $s2 = 2 * ((m + n) * m + m)$ |
| output gate | $s3 = (m + n) * m + m$ |
| memory unit | $0$ |
| LSTM | $s = Q_1 * 4 * ((m + n) * m + m)$ |
| CP | $s4 = 2mR$ |
| LSTM–CP | $S = Q_2 * 4 * ((m + n) * m + m) + 2mR$ |

**Table 2.** Network parameters.

| Structure | Parameter |
|-----------|-----------|
| forgetting gate | $s1 = 11,520$ |
| input gate | $s2 = 23,040$ |
| output gate | $s3 = 11,520$ |
| LSTM | $s = 92,160$ |
| CP | $s4 = 384$ |
| LSTM–CP | $S = 46,464$ |

The derivative of the function $f(x)$ at $x_0$ represents the slope of $y = f(x)$ at $x_0$, that is, the rate of change of $f(x)$ at $x_0$. The larger the derivative, the faster the change, that is, the faster the function grows. Because a variety of function variables that represent the parameter quantity occur in this article, drawing seems to be more difficult, such that the derivative is used for comparing multiple functions, as shown in Table 3.

**Table 3.** Parameter quantity derivative.

| Structure | Parametric Function Derivative |
|-----------|--------------------------------|
| LSTM | $S1'(Q) = 4 * ((m + n) * m + m)$ |
| LSTM–CP | $S2'(R) = 2m$ |

As can be seen from Table 3, when m and n are constant, the parameters of the LSTM network increase in a square form as the number of LSTM network layers Q increases, while the number of parameters in the BP neural network increases linearly when the number of neurons in the BP neural network (R) increases. Therefore, the LSTM–CP combination model can effectively reduce the number of parameters with the use of the approach presented in this paper.

## 3. Results

Yibin City is located in the southeastern part of Sichuan Province, China, with an area of 13,300 square kilometers. The city is located between $103°36'$–$105°20'$ east longitude and $27°50'$–$29°16'$ north latitude. Yibin is 298.7 km away from Chengdu, the capital of Sichuan Province in the north, and 583.5 km away from Kunming, the capital of Yunnan Province in the south, and the brief geographical location is shown in Figure 7. It is an important city from Sichuan to the middle and lower reaches of the Yangtze River and coastal areas. The terrain of Yibin City is dominated by hills and middle–low mountains, accounting for 91.9% of the city's total area. It belongs to a subtropical humid monsoon climate, and the annual average temperature is about 17.9 °C, the average temperature in January is 7.8 °C, and the average temperature in July is 26.8 °C. The water system of Yibin City is very complex and intertwined. The rivers in Yibin City are mainly the Yangtze River, the river network is dense, and the total water resources and hydropower resources are relatively abundant. The annual average precipitation is 1050–1618 mm, which is a typical humid area. The rainy season is concentrated in the summer and autumn. The precipitation in these two seasons accounts for 81.7% of the annual precipitation. The main flood season is

mainly July, August, and September. The precipitation in these three months accounts for about 51% of the annual precipitation.

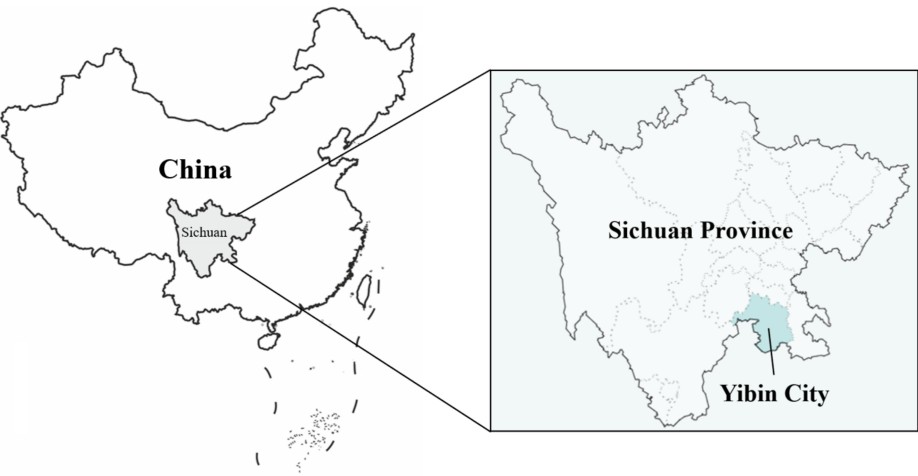

**Figure 7.** General location of Yibin City.

To eliminate the adverse effects of single sample data, improve the operation speed and accuracy as much as possible, and facilitate the operation of the model, it is necessary to first normalize the precipitation data [50] of Yibin City and map the original data to the interval [0, 1]:

$$x_i^{norm} = (x_i - x_{\min})/(x_{\max} - x_{\min}). \tag{24}$$

Mapping the input LSTM data to [0, 1] can aid in speeding up the convergence of the model. The use of CP for function fitting requires that the value of the data is supposed to be located in the interval [−1, 1], and the output data of LSTM will pass through the output gate, that is, through Equations (6) and (7), such that the output value of LSTM must be in the range of interval [−1, 1], meeting the requirements of the value range of the data fitted by the CP function.

In this paper, the common mean square error [53,54] is selected as the loss function of the training model, which is also adopted to calculate the validation set error of the model, and its formula is:

$$MSE = \frac{1}{n}\sum_{i=1}^{n}(x_i - \hat{x}_i)^2. \tag{25}$$

The Adam optimizer [55] has a fast convergence speed and can adjust the learning rate adaptively according to the data distribution, which is why the Adam optimizer is selected to optimize the error function in this paper. Additionally, Dropout [56] is added to the network to reduce the influence of over fitting [57,58].

The experimental environment adopted in this paper is as follows: the training platform is Windows 10 Home 64-bit operating system, the computer memory is 4G, the processor model is Intel(R) Core(TM) I5-6300HQ CPU @ 2.30ghz, and the graphics card model is NVIDIA GeFosrce GTX 960M 2G. With Anaconda as the development environment and Python3.7 as the programming language, the PyTorch 1.2.0 [59] deep learning framework is used as the development framework, and the Nvidia CUDA 10.0 computing platform is used for accelerated computing.

### 3.1. LSTM Parameter Setting

When LSTM is used to predict precipitation, network parameters of LSTM should be determined first. First, the learning rate [60,61] is fixed to 0.01 to determine the remaining parameters, and then the parameters including the number of LSTM network layers and the size of the hidden layer in the LSTM network will be changed. For LSTM networks

with different parameters, take the first NUM minimum errors, and the average prediction error is shown in Figure 8.

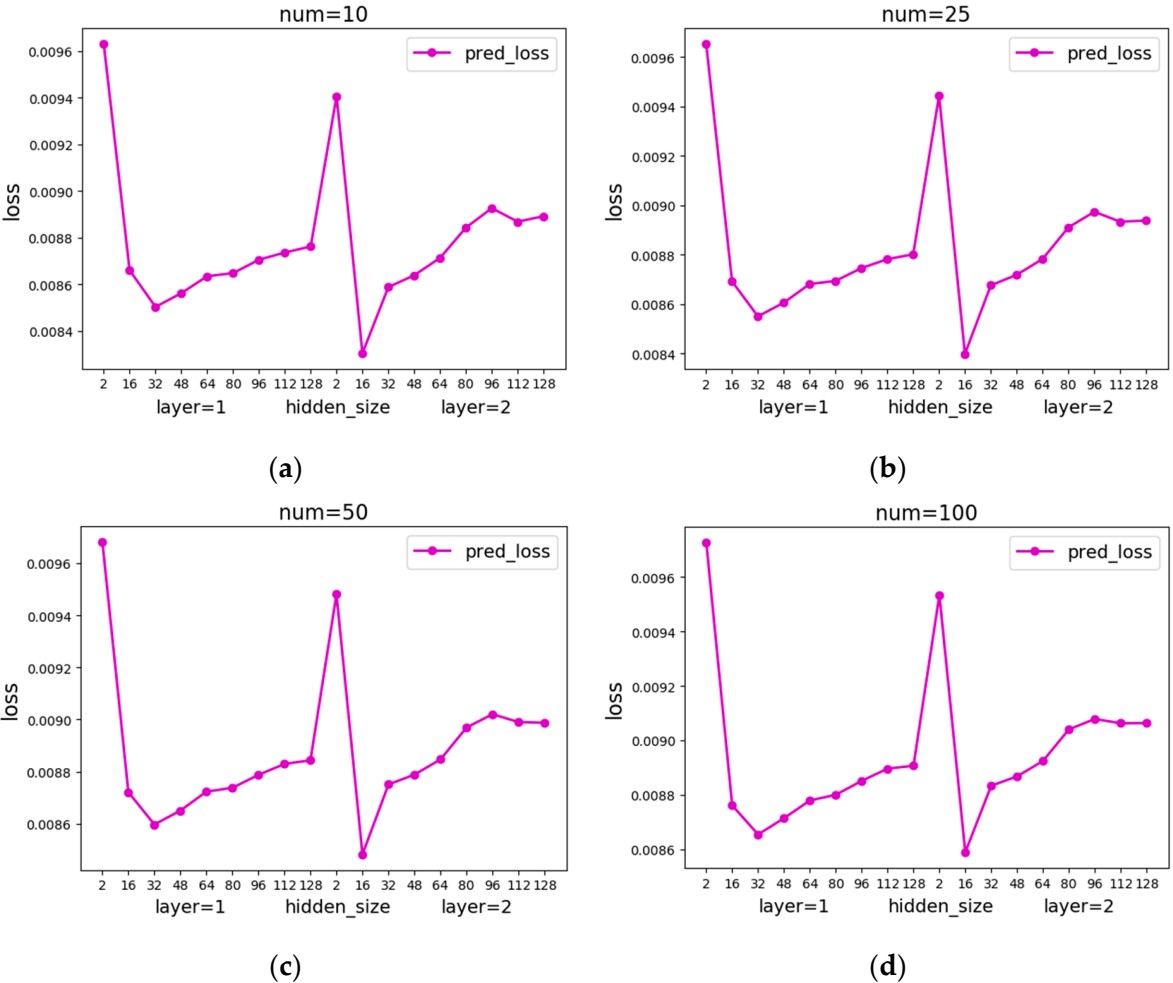

**Figure 8.** LSTM error curves of different layers and hidden_size. (**a**) Average error when Num is 10. (**b**) Average error when Num is 25. (**c**) Average error when Num is 50. (**d**) Average error when Num is 100.

It is not difficult to see from the error curve in Figure 8 that when layer = 2 and hidden_size = 16 of LSTM, loss reaches the minimum. Therefore, layer = 2 and hidden_size = 16 are taken in this paper. Then, test the learning rate, take layer = 2, hidden_size = 16, and take the learning rate 0.1, 0.05, and 0.01, respectively, for the experiment. As can be seen from Figure 9, the loss will eventually stabilize at 0.01 when the learning rate is 0.1, but the error will decrease slowly; when the learning rate is 0.05, the loss will eventually stabilize around 0.01, but it is not stable; when the learning rate is 0.01, loss quickly drops to 0.01 and remains stable all the time. In summary, this article sets the number of layers of the LSTM network to 2, the hidden features to 16, and the learning rate to 0.01.

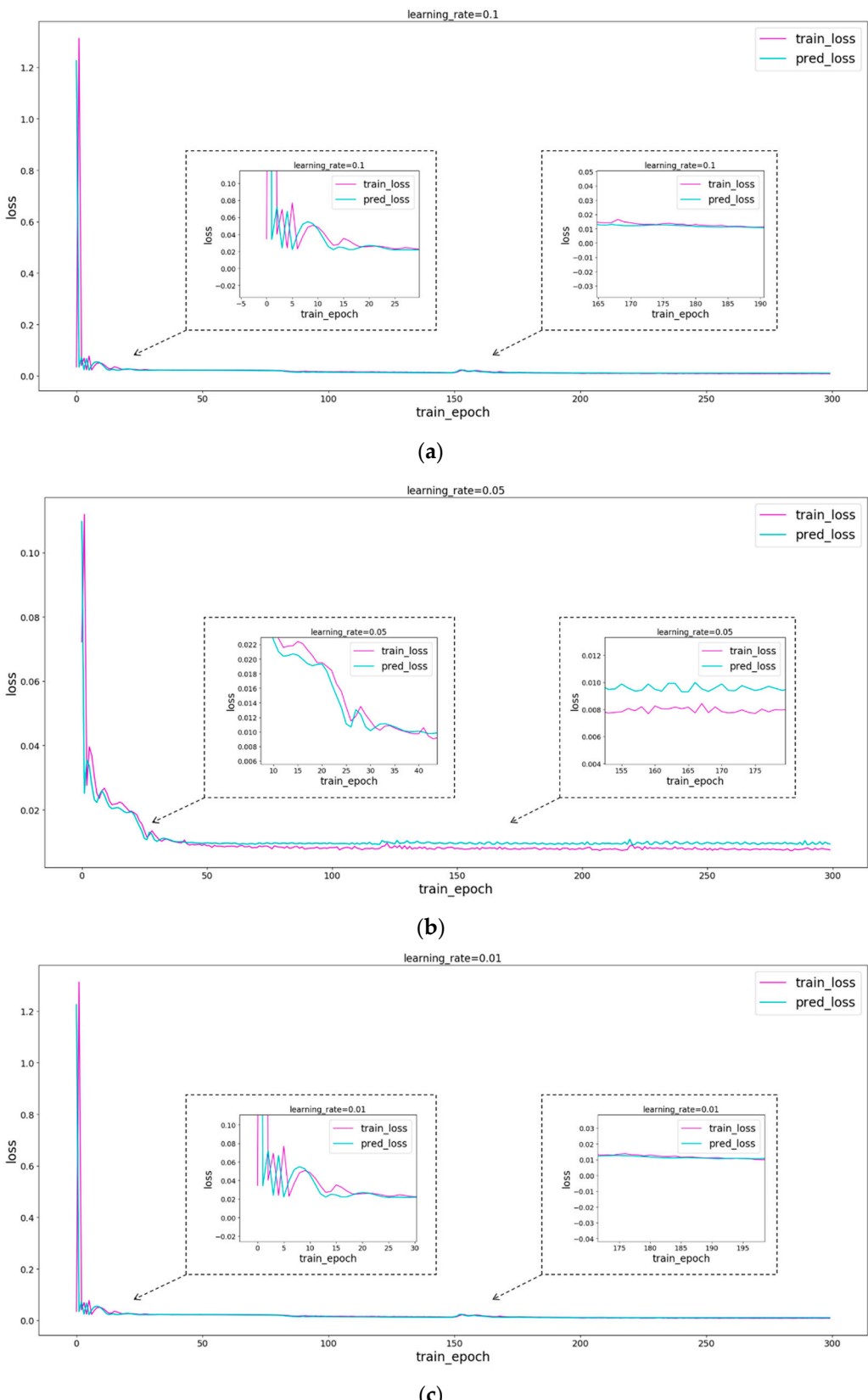

**Figure 9.** LSTM error curve for different learning rates. (**a**) LSTM error curve when the learning rate is 0.1. (**b**) LSTM error curve when the learning rate is 0.05. (**c**) LSTM error curve when the learning rate is 0.01.

### 3.2. LSTM–CP Parameter Setting

At the end of the LSTM network, there will be a linear layer, which can convert the time-series characteristic data extracted by the LSTM network into the target output [62,63]. The combination of CP and LSTM networks is supposed to use BP neural networks instead of this linear layer. With the Sigmoid function used as the common excitation function of the BP neural network, the Sigmoid function and CP function are, respectively, used as excitation functions to carry out comparative experiments in this paper. Figures 8 and 9 show that when layer = 2 and hidden_size = 16, the LSTM network performs the best. When layer = 1 and hidden_size = 32, the network is simpler but the model performance is relatively better. Therefore, this paper sets the basic LSTM network layer = 1, hidden_size = 32, learning_rate = 0.01 in the LSTM–CP combination model.

The next step is to determine the number of neurons in the BP neural network. Zhang Y proposed a two-stage approach to determine the number of neurons. In the first stage, the number of neurons is increased to a large extent to determine the approximate value range of neurons. In the second stage, the number of neurons is increased one by one to determine the exact value of the number of neurons. This method can effectively determine the number of neurons in the BP neural network. In order to obtain the approximate value range of neurons quickly and determine the number of neurons accurately, the initial number of neurons is set to 5 in the first stage of this paper, with a step size of 5 to increase neurons. First, Sigmoid is tested as the excitation function, and the prediction error of the minimum NUM among the error values of different neurons is taken as the average error curve.

It can be seen from Figure 10 that the error fluctuates as the number of neurons changes. When the number of neurons is about 70, the error is small, which means that the loss of the network is relatively small and stable when the number of neurons is about 70, and the optimal number of neurons is about 70. Therefore, the number of neurons is set from 66 to 74 for the experiment. As can be seen from Figure 11, when the number of neurons is 67, the prediction error is the smallest, that is, when the Sigmoid function is used as the excitation function of the BP neural network, the optimal number of neurons is 67.

Then, it is necessary to determine how many neurons should be used as the excitation function of the BP neural network, and the minimum NUM prediction errors among the error values of different numbers of neurons are still taken as the average error curve. The first-order CP transforms all the input into 1, while the second-order CP is actually a linear function. Since the first-order and second-order CP do not have nonlinear characteristics, they are not suitable for excitation functions of neural networks, so the CP order is at least 3 in this paper. A good fitting effect can be obtained at a lower order due to CP's strong fitting ability, which makes it unnecessary to use the two-stage method to determine the order of CP, and the order can be increased from 3 to 3. Figure 12 shows the experimental results of CP with different orders as the excitation function. It is not difficult to see that the error is low when the CP order is 6. Therefore, this paper sets the CP order of LSTM–CP to 6, that is, the number of neurons in the hidden layer of the BP neural network is 6.

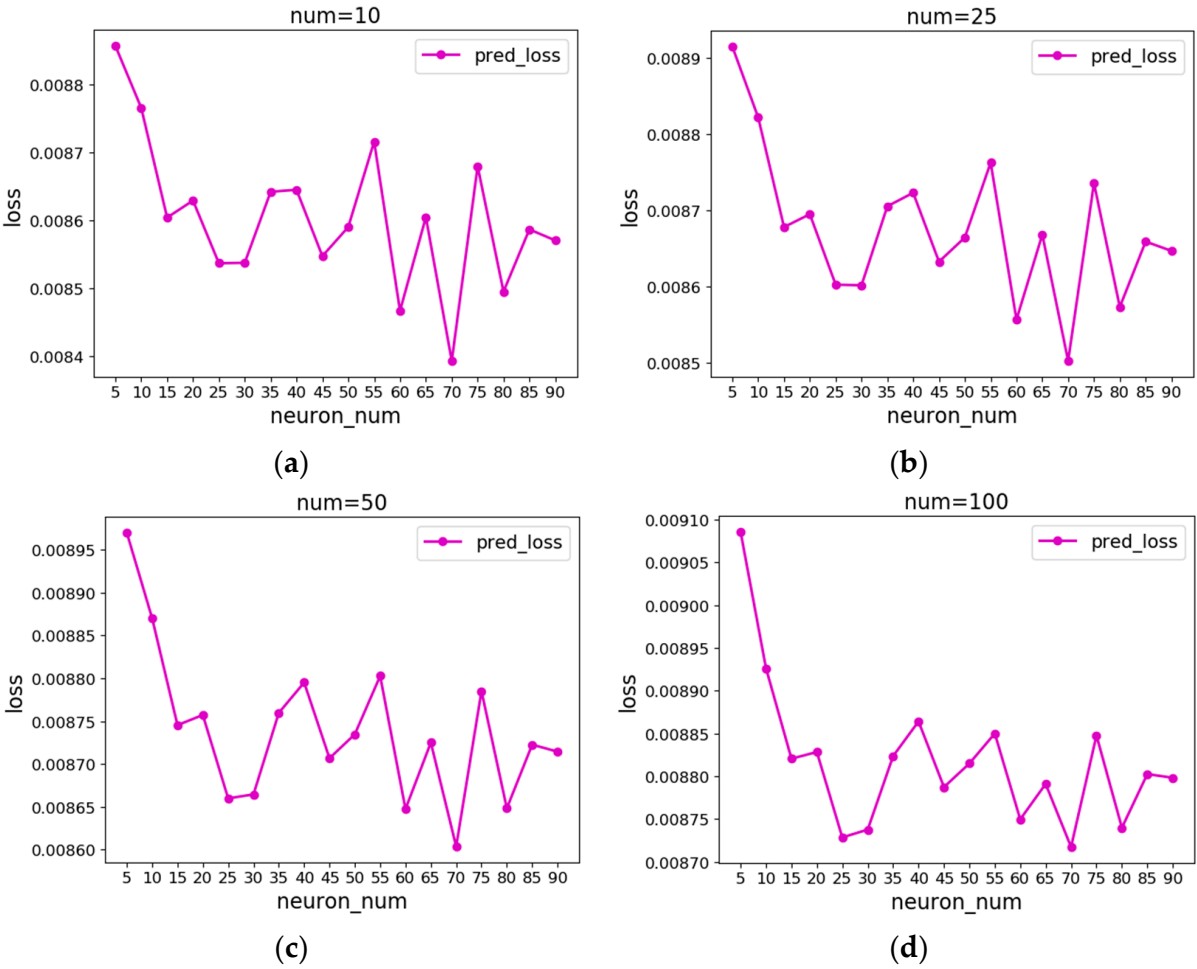

**Figure 10.** LSTM–BP (Sigmoid): the first-stage error curve. (**a**) Average error curve when Num is 10. (**b**) Average error curve when Num is 25. (**c**) Average error curve when Num is 50. (**d**) Average error curve when Num is 100.

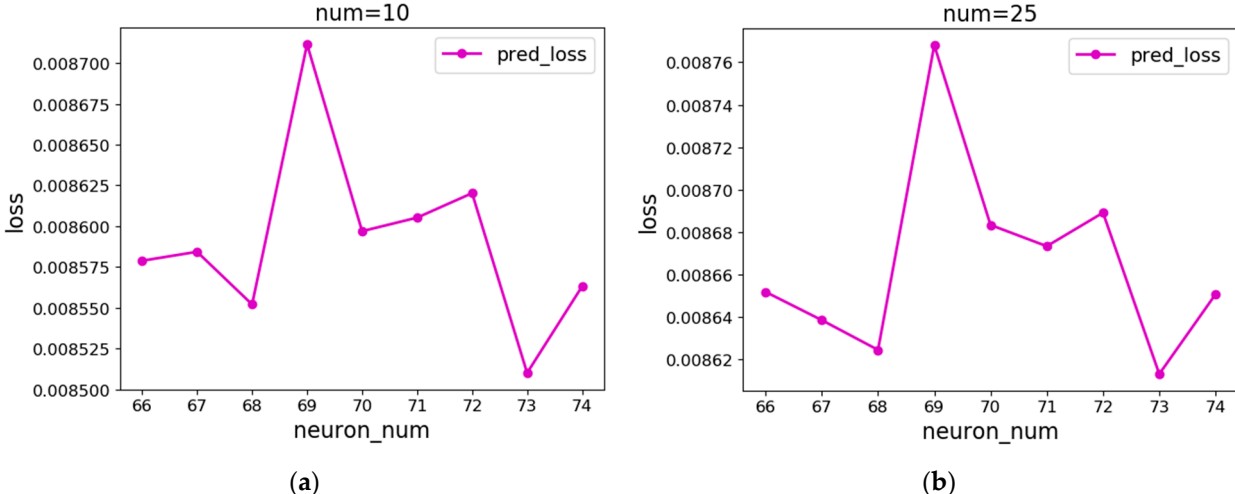

**Figure 11.** *Cont*.

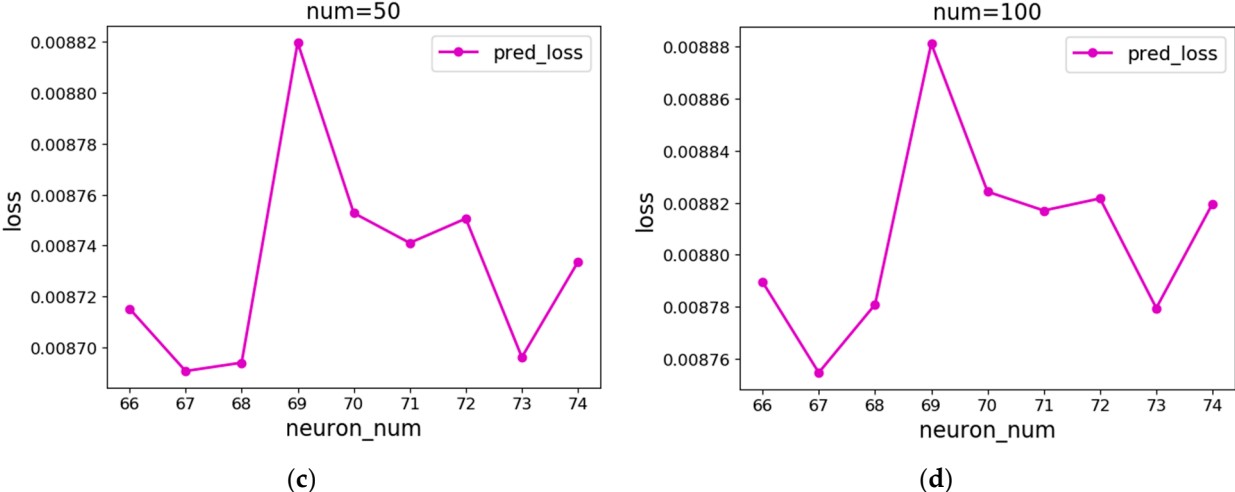

**Figure 11.** LSTM–BP (Sigmoid): the second-stage error curve. (**a**) Average error curve when Num is 10. (**b**) Average error curve when Num is 25. (**c**) Average error curve when Num is 50. (**d**) Average error curve when Num is 100.

**Figure 12.** LSTM–CP: Error curve. (**a**) Average error curve when Num is 10. (**b**) Average error curve when Num is 25. (**c**) Average error curve when Num is 50. (**d**) Average error curve when Num is 100.

*3.3. Comparative Analysis*

The optimal parameters of different networks are given in Section 3.1, while this section will make a detailed comparative analysis of the performance of each model. Figure 13 shows the error curves of different models, and Figure 14 shows the prediction results of different models. It can be easily seen from the error in Figure 13 that the LSTM network has a stable error of around 0.01 after 100 rounds of training, LSTM–CP has a stable error of around 0.01 after 100 rounds of training, and LSTM–BP (Sigmoid) also has a stable error of around 0.01 after 100 rounds of training. The reason for this is that the LSTM network has a strong ability to process sequence data and can quickly extract sequence features. We ran each model separately and obtained the prediction results of different models, as shown in Figure 14. It is not difficult to see from Figure 14 that the prediction results obtained by all networks are very close to the original data.

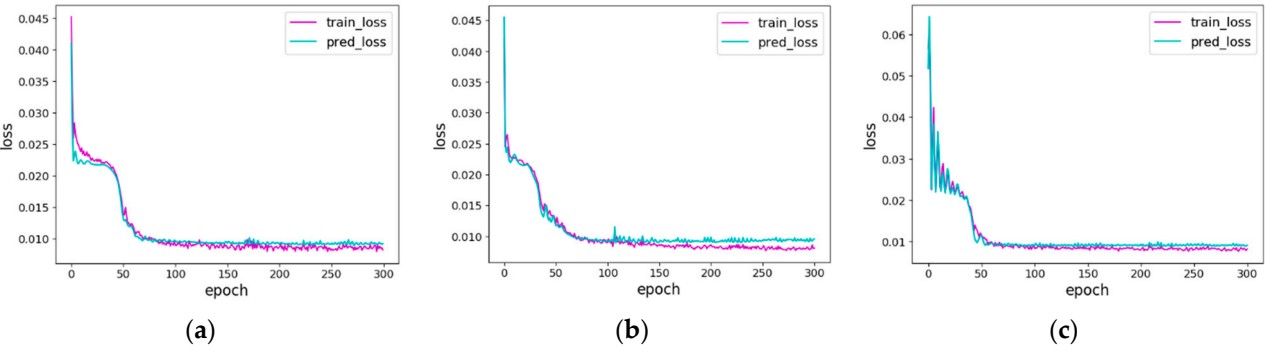

**Figure 13.** Training and prediction error curves of different networks. (**a**) LSTM. (**b**) LSTM–CP. (**c**) LSTM–BP (Sigmoid).

According to Figures 13 and 14, each model has better performance. Listing the detailed data of each model in Table 4, it is not difficult to see that the training error, prediction error, and training time of LSTM–CP are less than those of an ordinary LSTM network. In particular, if the excitation function of the BP neural network is operated on the CPU, the running time of CP will be shorter than that of Sigmoid. Therefore, using CP as the excitation function can obtain the lowest training error, the prediction error is smaller, and the running speed is better than the LSTM network.

**Table 4.** Comparison of model effects.

| Model | Training Error | Prediction Error | Running Speed |
|---|---|---|---|
| LSTM | 0.0078 | 0.0091 | 4.95 |
| LSTM–BP (Sigmoid) | 0.0079 | 0.0090 | 3.19 |
| LSTM–CP | 0.0076 | 0.0090 | 4.62 |

Note: The running speed is s/100 times.

Next, by using the ARMA linear model, SVR model, and MLP model to predict the precipitation at the same time, we calculated the evaluation indexes of each model, and compared the results with the LSTM–CP model proposed in this paper, as shown in Table 5. It is not difficult to see that, compared with other models, the values of MAE (mean absolute error), MSE (mean square error), and MAPE (mean percentage error) of the LSTM–CP network model are smaller than other models, which indicates that the LSTM–CP network model proposed in this paper has higher consistency and accuracy in rainfall prediction.

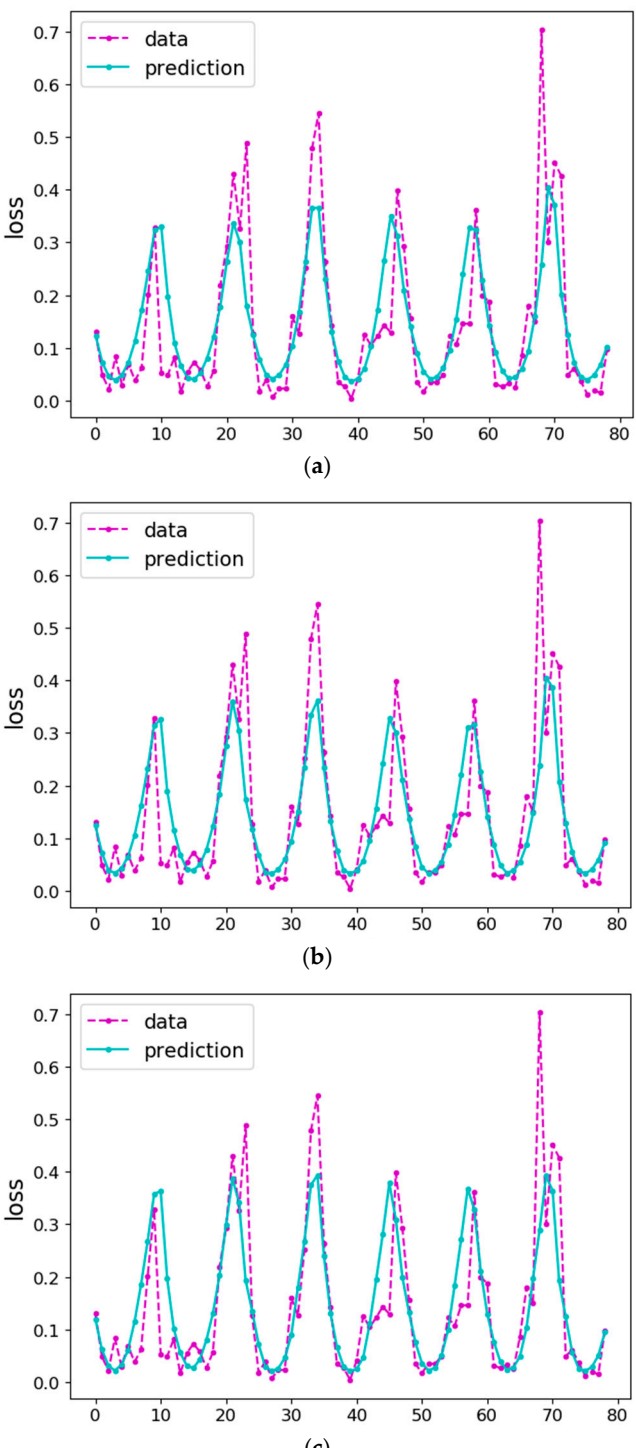

**Figure 14.** Forecast results. (**a**) LSTM. (**b**) LSTM–CP. (**c**) LSTM–BP (Sigmoid).

**Table 5.** Comparison results of prediction models.

| Model | MAE | MSE | MAPE |
|---|---|---|---|
| ARMIA | 0.0836 | 0.0120 | 55.051 |
| SVR | 0.0925 | 0.0172 | 65.731 |
| MLP | 0.1101 | 0.0191 | 75.210 |
| LSTM–CP | 0.0601 | 0.0090 | 53.121 |

## 4. Discussion

Due to the complex and diverse causes of precipitation and the interaction of various factors [64], it is very difficult to establish a mathematical model [65] of precipitation. Deep learning can automatically extract the low-level features of data and form abstract high-level features, without the need for the physical modeling of data, and it can easily deal with complex data structures because of its strong nonlinear ability [66]. The LSTM network is often employed to process time-series data. Its strong memory ability makes it have natural advantages in processing time-series data.

As can be seen from (a) in Figure 14, the precipitation value predicted by the LSTM neural network model is basically consistent with the real value of precipitation data, and LSTM can accurately extract the time-series features hidden in precipitation data. However, the LSTM network has some problems such as complex structure and gradient disappearance. Therefore, this paper proposes the LSTM–CP combination model by combining LSTM and CP, which makes full use of LSTM's ability to predict series data and CP's powerful function fitting ability in order to ensure the accuracy of the model, reduce the parameters of the network, and reduce the complexity of the precipitation prediction model.

Table 2 shows that using the LSTM–CP combination model can effectively reduce the number of network parameters. The amount of LSTM network model parameters is 92160, while the amount of LSTM–CP combination model parameters is only 46464, which greatly reduces the complexity of the model and is suitable for processing large and medium-sized datasets. At the same time, Table 4 shows that compared with the single LSTM model and the traditional precipitation prediction model, the LSTM–CP combined model has a smaller training error and test error, higher prediction accuracy, and is more suitable for precipitation prediction research. Because the model can reduce the use of parameters to a higher degree, it can more effectively reduce the running time when dealing with large and medium datasets, and make the processing of sequence data more efficient.

Rainfall is affected by the fluctuations of sea and land locations, topography, latitude, and human factors, but in this study, we ignored these changes. In future research, LSTM–CP can be applied to the scene with complex and huge data, such as text, music, and other sequence data processing. In this case, the number of network layers and hidden layers is larger when LSTM is used alone, and the combination of LSTM and CP can make the parameters have a larger space to decline, and it is not easy to overfit. In addition, the derivative function can be determined in advance according to the order of CP without using the deep learning framework for automatic derivation, which improves the computational efficiency.

## 5. Conclusions

Natural disasters often lead to major and long-term damage to the entire socio-economic system, such as floods, which may damage multiple infrastructure systems, lead to cascading failures and major socio-economic losses, and hinder development. Therefore, reducing the risk of precipitation disaster is closely related to sustainable development. With the progress of technology in recent years, artificial intelligence has become the main driver in various fields including sustainable development. Deep learning improves the ability to deal with complexity and increases our understanding of the variables and sources that affect rainfall. This paper proposed the LSTM–CP model to predict the precipitation of Yibin City. Firstly, the BP neural network is combined with LSTM to form a combined model where the LSTM network is used to extract the sequence features of the precipitation data. Then, the BP neural network is used to process the sequence features to obtain the target output. Because the traditional BP neural network has the disadvantages of easily falling into local minimums and saddle points, this article considers using CP as the excitation function to replace the Sigmoid function in the BP neural network, with the powerful function fitting ability of CP to process sequence features.

Through experimental tests and comparative analysis, the LSTM–CP combination model proposed in this paper has fewer parameters, a shorter running time, and smaller prediction error than the LSTM network. At the same time, compared with the SVR model, ARIMA model, and MLP model, the prediction accuracy of the LSTM–CP combined model is significantly improved, which improves the accuracy of rainfall prediction and makes the model more applicable. It can reflect the change trend of precipitation and help provide a data reference in areas prone to floods and drought disasters to help relevant departments prepare in advance, reduce local economic losses, and better achieve sustainable development. Furthermore, the rainfall prediction model can be incorporated into the regional early warning system to help better plan and manage water resources and reduce the risk of flooding. Finally, the application of artificial intelligence to precipitation prediction provides new ideas and methods for the current precipitation prediction research, and opens up a broader space for realizing the goal of sustainable development.

**Author Contributions:** Y.G. conducted the experiments and the whole article. W.T. collected and sorted out the data, and revised and improved the article. G.H. constructed the framework of the whole paper and wrote the review. F.P. designed the experiment and methodology. Y.W. wrote the original draft preparation. W.W. provided formal analysis and experimental tools. All authors have read and agreed to the published version of the manuscript.

**Funding:** This research was supported by the Key Laboratory of Agricultural Information Engineering of Sichuan Province and Social Science Foundation of Sichuan Province in 2019, grant number SC19C032.

**Institutional Review Board Statement:** Not applicable.

**Informed Consent Statement:** Not applicable.

**Data Availability Statement:** The data used in the research of this paper comes from the website: http://dataju.cn/Dataju/web/home (accessed on 15 October 2021).

**Acknowledgments:** Thanks for the help of the Key Laboratory of Agricultural Information Engineering of Sichuan Province.

**Conflicts of Interest:** The authors declare no conflict of interest. The funders had no role in the design of the study; in the collection, analyses, or interpretation of data; in the writing of the manuscript, or in the decision to publish the results.

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
