# Peer review of "Research on Precipitation Forecast Based on LSTM–CP Combined Model"

_sustainability, doi:10.3390/su132111596_

Round 1

Reviewer 1 Report

The manuscript by Guo et al. has clearly improved compared to the previous submission, in structure, content and readability. However, I believe it can be further improved, with a modest effort from the authors. To this end, I would ask the authors to appropriately address the following comments:

  • The authors should better highlight in the introduction the advantages of the proposed approach compared to more classic machine learning algorithms in solving hydrological problems.
  • The authors should explain why normalizing the input data, in this case, leads to better results than other pre-processing operations, such as standardization.
  • The authors should provide compelling explanations as to why the model's predictions always underestimate the highest rainfall values.
  • The authors could draw useful insights from other recent works based on the application of LSTM for the prediction of hydrological quantities. To this end, I would advise the authors to consider and include among the references the following articles:

Ferreira, L. B., & da Cunha, F. F. (2020). Multi-step ahead forecasting of daily reference evapotranspiration using deep learning. Computers and Electronics in Agriculture178, 105728.

Granata, F., & Di Nunno, F. (2021). Forecasting evapotranspiration in different climates using ensembles of recurrent neural networks. Agricultural Water Management255, 107040.

Kratzert, F., Klotz, D., Brenner, C., Schulz, K., & Herrnegger, M. (2018). Rainfall–runoff modelling using long short-term memory (LSTM) networks. Hydrology and Earth System Sciences22(11), 6005-6022.

Xiang, Z., Yan, J., & Demir, I. (2020). A rainfall‐runoff model with LSTM‐based sequence‐to‐sequence learning. Water resources research56(1), e2019WR025326.

Based on previous comments, I think the article needs to undergo moderate revisions.

Reviewer 2 Report

The paper has significantly improved.

Author Response

Thank you very much for your recognition of our work. In the revised manuscript, we added some references. We will work harder and strive for greater achievements in the future.

Sincerely,

Yan Guo on behalf of the authors

Round 2

Reviewer 1 Report

The article has been further improved and is ready to be published. Just a comment: it is advisable to check the format of the references, both in the body of the article and in the list, since there are inaccuracies.

This manuscript is a resubmission of an earlier submission. The following is a list of the peer review reports and author responses from that submission.

Round 1

Reviewer 1 Report

The paper by Guo et al. presents a combined model formed by Long Short-Term Memory (LSTM) network and Chebyshev polynomial (CP) to forecast precipitation in Yibin City, China.

After a long description of the technical aspects of the forecasting model, the authors state that its accuracy is significantly higher than that of similar forecasting models based on SVR, MLP, and ARIMA.

The idea is valuable, and the model shows interesting potential. However, the article has significant flaws.

Since these defects are of a structural nature, I will limit myself to general comments, without going into details.

The manuscript is unbalanced. Much of the article is devoted to the description of the technical features of the model, most of which is already known from the literature. The sections of the results must certainly be expanded and argued in more detail. Presentation and discussion of results are not effective in the current form of the article. The authors are reminded that Sustainability is a Journal of environmental, cultural, economic, and social sustainability of human beings. The article should be better placed within the scope of the Journal.

Furthermore, it is not at all clear how the graphs, which are almost all of low quality, allow us to understand the results.

The comparative analysis is not convincing at all. The authors do not explain how the SVR, MLP, and ARIMA-based models were optimized. On line 474 you can read: “The optimal parameters of different networks are given in sections 4.1 and 4.2”, but sections 4.1 and 4.2 are missing. Without adequate optimization of the other models, any comparative analysis is meaningless, and the results do not justify the conclusions.

English needs to be revised by a native speaker. There are several typos.

Reviewer 2 Report

The survey is very interesting and also the methodology you have prepared seems to work very well in relation to other methods with which you have compared it. To improve the discussion where there is no reference to existing literature.

Reviewer 3 Report

The manuscript “Research on Precipitation Forecast Based on LSTM-CP Combined Model” proposes an alternate approach for precipitation forecasting by adopting Chebyshev polynomial (CP) with LSTM and BP to reduce the computational time and overcome some limitations of these techniques such as local minima. The manuscript is not clear and difficult to read. The language is very poor making it more difficult to follow. Here are some recommendations for improvements and questions. 

  • Introduction section does not satisfactorily justify the need and motivation of the work. Every paragraph is disconnected making it difficult to read and there are several bold statements. The section can be improved by considering the following points:
  1.     Discuss what kind of data is used and its importance. Is it satellite data or surface rain gauge observations? 
  2.     Describe study area and provide a figure
  3.     What kind of precipitation events are under consideration? For example, forecasting extreme rainfall? 
  4.     There are several other methods of forecasting than regression such as cloud motion tracking using satellite observations and Langrangian techniques. Few examples listed here: 

https://doi.org/10.1016/j.jhydrol.2003.11.011

https://doi.org/10.1016/j.atmosres.2012.07.001 

  1.     What is the scale of forecasting like daily? Seasonal? Hourly? Sub-Hourly? The significance of the work changes based on this and should be discussed in Introduction. 
  2.     The limitation discussed is not satisfactory. For example, Line 94-106 needs support/references for the claim authors are making. 
  3.     Author’s do not discuss anything relevant to Convolutional Neural Networks as a feature extraction. Why should only LSTM be used? 
  4.     CP is suddenly introduced without any background of what extinction functions are.
  • Section 2: Provide information of dataset, study area, study period etc first before describing the method.
  • Section 2.1.: Several things are already reported in Introduction and it is repeated here and in discussion and conclusion sections making it redundant. 
  • Section 2.2: This section looks more like a report on LSTM networks which is well known to the community with several textbooks and articles. Therefore it is strongly recommended  to condense this section by giving focus on other details of the study.
  • Line 248-251: Deeper networks are considered to be more efficient than regular shallow NN. Why did authors restrict on using only one hidden layer?
  • Line 276-278: There are solutions for this such as controlling the learning rate. The statement is too bold.
  • Line 279: Depending on the application requirement there are several other activation functions exits such as linear, exponential etc.
  • Line 296-301: Redundant
  • Line 354-361: This whole paragraph is very confusing and difficult to follow. For example, 
  1.     Why are the authors citing something in Section 4.1. now itself? 
  2.     How did authors come up with these numbers given here?
  3.     If the 90% samples are randomly selected, then will it hold the temporal relationship? 
  4.     If 90% of data is used for training, then only 10% is used for validation? Many things are unclear.
  • Line 378-378: Can you provide a justification why it is important to normalize when the data is large? Generally, normalization is performed if there are more than one predictor which are in different data ranges/extent.
  • Line 406: What does “num” stand for?
  • Table 4: 
  1.     What are the running speed units here? 
  2.     Are these values statistically different? They look very close to each other. 
  3.     Also, the validation data should be independent of test data which is used to tune/select the model parameters.
  • Table 5: 
  1.     Again, what are the units here? 
  2.     Are these statistically different? A significance test is required to check this. It is highly recommended to use independent data for validation.
  • Line 516-517: The current paper results do not show significant difference in the performance.